# Nutrition and Sarcopenia—What Do We Know?

**DOI:** 10.3390/nu12061755

**Published:** 2020-06-11

**Authors:** Aravinda Ganapathy, Jeri W. Nieves

**Affiliations:** 1Institute of Human Nutrition, Columbia University, New York, NY 10032, USA; ag4189@cumc.columbia.edu; 2Epidemiology, Mailman School of Public Health, Columbia University, New York, NY 10032, USA; 3Hospital for Special Surgery, New York, NY 10021, USA

**Keywords:** sarcopenia, protein, vitamins, minerals, foods, diet

## Abstract

Muscle health is important for the functionality and independence of older adults, and certain nutrients as well as dietary patterns have been shown to offer protective effects against declines in strength and function associated with aging. In this paper, micronutrients, macronutrients, and food groups have been reviewed, along with their studied effects on the prevalence and incidence of sarcopenia, as well as their ability to preserve muscle mass and optimize physical performance. Randomized controlled trials appear to suggest a critical role for dietary intake of protein in preventing sarcopenia and muscle loss, although the optimal dose and type of protein is unknown. There are some promising data regarding the role of vitamin D and sarcopenia, but it is unclear whether the dose, frequency of dose, or length of treatment impacts the efficacy of vitamin D on improving muscle mass or function. Selenium, magnesium, and omega 3 fatty acids have been studied as supplements in clinical trials and in the diet, and they appear to demonstrate a potential association with physical activity and muscle performance in older individuals. Following the Mediterranean diet and higher consumption of fruits and vegetables have been associated with improved physical performance and protection against muscle wasting, sarcopenia, and frailty.

## 1. Introduction

Sarcopenia consists of a loss of skeletal muscle mass and physical function (muscle strength or physical performance) that occurs with advancing age and has become increasingly prevalent [1,2,3]. Sarcopenia causes concern because it may result in many adverse outcomes for older adults, including physical disability, poor quality of life, and increased mortality [2]. Sarcopenia is responsible for considerable healthcare expenditure, with direct medical costs attributable to the disorder estimated at US $18.5 billion in the United States in 2000 [4]. Sarcopenia generally refers to class II sarcopenia, which is defined as individuals with an appendicular skeletal muscle mass (ASM)/height^2 ratio below 2 SD. Class I sarcopenia is defined as individuals with an ASM/ht^2 ratio between 1 and 2 SD [5]. Sarcopenia is often assessed by physical measurements of whole-body lean mass, appendicular lean mass, or muscle mass and size. Alternatively, physical performance is assessed as a proxy for sarcopenia using grip strength, leg muscle strength, walk speed, or a battery of physical performance (walk speed, chair stands, and balance). Certain risk factors for sarcopenia have been previously explored, such as smoking, diabetes, medications, and BMI [6,7,8]. However, suboptimal diets and poor nutritional status are common in the elderly, particularly in frail individuals, and therefore improving diet and nutrition may be effective for both the prevention and treatment of sarcopenia [9]. Although muscle atrophy occurs with aging, the fact that there is a variability in this process [10] indicates that there are potential factors that might influence the rate of decline of muscle strength and function. Significant physiological changes occur with aging which may be linked to the loss of muscle and development of sarcopenia; two of these that have been recently investigated are an increased level of reactive oxygen species (ROS), and systemic chronic low-grade inflammation [11]. In a study of 12 countries, almost two-thirds of older study participants were identified as being at nutritional risk or malnourished [12]. These lower intakes of food lead to weight loss and potential muscle loss. The deficiencies in the overall diet and/or specific nutrients that result from lower intake of food may relate to sarcopenia. Given that nutrition may influence the development of sarcopenia, a common condition of aging, this topic deserves further review. This review will include studies in which sarcopenia was evaluated as loss of muscle mass, (dual x-ray absorptiometry (DXA), magnetic resonance imaging (MRI), computerized tomography (CT), anthropometry, or bio-electric impedance) muscle strength, (grip strength, isometric, isokinetic) or physical function tests.

## 2. What We Know About Nutrition and Sarcopenia

### 2.1. Macronutrients

#### Protein

Dietary protein provides amino acids that are needed for the synthesis of muscle protein. It is unclear whether protein requirements need to be higher in older adults in order to maintain nitrogen balance and prevent loss of muscle mass and strength [13]. There have been several randomized controlled trials to evaluate the role of protein supplementation alone (Table 1) or in combination with other nutrients (Section 2.5). In addition, it is possible that protein supplementation should be combined with exercise to prevent sarcopenia, and this was recently reviewed [14].

Several cohort studies have found an association between sarcopenia and protein such that lower protein intakes are related to a loss of lean mass by DXA [23,24] and to a reduced grip strength [25,26].

Randomized controlled trials have predominantly used varying doses of whey protein and in several instances combined it with exercise. Kang [17] provided 32.4 g of whey protein vs. a control diet for 12 weeks along with a 30 min home-based resistance exercise program taught by a professional physical therapist to 115 male and female subjects over the age of 60 with some markers of frailty. At the end of the 12 weeks, the whey supplemented group showed a significant increase in grip strength, gait speed, and time to complete chair stands. In another study, 120 subjects aged 70 to 85 with mild frailty were given whey protein at doses of 0.8, 1.2, and 1.5 g/kg body weight for 12 weeks. The highest versus lowest dose of whey protein supplementation led to a significant increase in appendicular skeletal muscle mass, skeletal muscle mass index and gait speed [18]. In another study, 112 people over the age of 65 with sarcopenia or dynapenia, supplementation for 12 weeks was given as 10 g whey protein along with 800 IU vitamin D with or without a program of resistance exercise as compared to exercise alone and a control group. In this study, the combination of exercise and whey protein improved appendicular muscle mass in sarcopenic adults, but not those with normal muscle mass but reduced muscle strength [19]. In a smaller study of 28 people over the age of 70 years who completed the 12 week study, 20 g of whey protein isolate and 3 g of leucine as compared to the placebo group, showed no improvement in physical function or muscle mass [20]. This study was limited by the very small sample size. In a study of 80 people (aged 70–85), whey protein concentrate (40 g/day), in combination with progressive high-intensity resistance training, did not result in statistically significant differences in lean mass, muscle cross sectional area or stair-climbing performance, when compared to the control group [21].

In two older studies a milk-based supplement was used. In a study of 218 men and women aged 75–96 with indications of sarcopenia or dynapenia, 40 g of milk protein versus an isocaloric placebo for 12 months led to no significant differences in muscle mass or physical performance [15]. In a group of 62 pre-frail or frail individuals over the age of 65, 15 g of a milk protein concentrate versus an isocaloric placebo were provided for 24 weeks. In this study, there was an increase in lean body mass with supplementation as compared to exercise alone, but no effect on strength or physical performance was observed [16]. In another study, lean red meat was provided to increase dietary protein intake and this was combined with resistance exercise training [22] and in 100 women treated for 4 months, lean tissue mass and muscle strength were improved with supplementation. 

Branched chain amino acids have also been studied. A systematic review and meta-analysis of leucine concluded that leucine increases muscle protein synthesis in older individuals, and may be of benefit to address age-related declines in muscle mass [27]. An earlier review has also discussed several other studies regarding protein and sarcopenia [28]

In frail elderly individuals it may be that a combination of exercise with additional protein intake may help to minimize the loss of lean mass and diminished strength that occurs with aging. Whether protein needs to be supplemented or if the recommended intakes of protein should be increased [29] is unclear. Further trials are needed to see what amount and what source of dietary protein may provide the best benefit to prevent sarcopenia.

### 2.2. Vitamins

#### 2.2.1. Vitamin D

Vitamin D supplementation (Table 2) has been studied in both vitamin D deficient [30,31] and community populations [32], and in those with [30] or without sarcopenia [31,32]. In a study of 118 people with both sarcopenia and vitamin D deficiency, a nutritional supplement of 10,000 IU given 3 times a week vs. the nutritional supplement alone led to an improvement in muscle mass but not grip strength after 6 months [30]. It is also possible that obesity may modify the effect of vitamin D on sarcopenia, as a stronger effect size for vitamin D supplementation was seen in normal weight subjects vs. obese subjects [30]. Furthermore, while gender was a significant predictor of muscle mass in normal weight subjects, it was not a significant predictor in obese patients [30]. In a trial of 88 vitamin D deficient younger (age 45 to 60) women, 40,000 IU ergocalciferol once a week in addition to a nutritional supplement after 3 months did not change muscle mass, muscle strength or size measured by bioelectrical impedance, hand-grip strength test, or ultrasound [31]. In postmenopausal women (n = 160) aged 50 to 65 (with average serum 25(OH)D levels around 15 ng/mL), 1000 IU of daily cholecalciferol for 9 months led to a differential increase in the muscle strength of lower limbs in the vitamin D group [32]. It is unclear whether the dose, frequency of dose, or length of treatment impacts the efficacy of vitamin D on improving muscle mass or function.

#### 2.2.2. Vitamin C

One of the factors potentially implicated in the mechanism of sarcopenia is oxidative stress [33]. Reactive oxygen species (ROS) can trigger atrophy and the loss of muscle function directly, as well as upregulate inflammatory cytokine expression such as TNF, IL-6, and IL-1 [33,34]. As a result, antioxidants have been suggested to combat the development of sarcopenia by inhibiting the generation of ROS. Vitamin C plays a major role in the regeneration of vitamin E in cell membranes, and acts to reduce the vitamin E radicals generated; as a result, it is considered the most important hydrophilic antioxidant [35]. 

Studies have shown mixed results overall regarding whether vitamin C dietary intake is related to muscle strength or physical performance. In the New Mexico Aging Process Study, only women with a slower gait speed demonstrated lower vitamin C intake, while men with a slower gait did not [36]. In the Hertfordshire Cohort Study, Vitamin C intakes were associated with faster chair-rise times and 3 m walk times in women, but not in men [37]. These studies appear to suggest a potential sexual dimorphism regarding antioxidant intakes, especially vitamin C, with muscle mass and physical performance. However, another large study involving 1433 subjects (658 men and 775 women) over the age of 60 years found no association between vitamin C and sarcopenia in either sex, as assessed by appendicular skeletal mass divided by weight being less than 1 SD below the mean of a reference sample [38]. It is important to note that certain studies have shown associations between vitamin C and physical performance as well: an analysis of the InCHIANTI study demonstrated that daily dietary vitamin C intake was significantly associated with both knee extension strength and overall physical performance (based on walking speed, a chair-stand test, and a balance test) [39]. Another study of older adults in Spain demonstrated a significant inverse association between vitamin C dietary intake and incident frailty [40]. High doses of antioxidant supplementation (vitamin C + E) have actually been shown to constrain total lean mass gains in a study involving 34 elderly males randomized to either an antioxidant or placebo group [41]; however, this supplementation has yet to be tested on women. It is unclear whether vitamin C dietary intake is beneficial for decreasing sarcopenia, and whether there is truly a sexually dimorphic effect. Overall, the effect of vitamin C dietary intakes and supplementation should be further studied in both men and women. 

#### 2.2.3. B-Vitamins 

Pyridoxine, or vitamin B_6_, is naturally abundant in meats and poultry [42]. Clinical symptoms of marked deficiency are largely neurological, and can in turn affect motor neurons, leading to weakness and loss of distal sensation [43]. However, the effects of chronic suboptimal dietary intake in elderly individuals is far less understood. In a cross-sectional analysis of 227 individuals from the Maastricht Sarcopenia Study, researchers found that sarcopenic older adults consumed significantly less vitamin B6 than non-sarcopenic adults [44]. Another cross-sectional analysis conducted on Dutch older adults found that greater intakes of vitamin B6 were partially correlated with higher scores on the Short Physical Performance Battery (SPPB) and the chair-rise test [45]. A link between frailty and B6 has been established as well: one cross-sectional study measuring serum concentrations of micronutrients found that frail women had significantly lower serum levels of vitamin B6 than did non-frail or prefrail women [46]. A prospective cohort study of 1643 individuals designed to evaluate the relationship between dietary intake and incident frailty found that nonadherence to the RDA of vitamin B6 was associated with higher odds of frailty after 3.5 years of follow up [40]. In a randomized controlled trial conducted on rats, it was discovered that even compared to marginal B6 deficiency, consuming the RDA of vitamin B6 upregulates the expression of several protective genes such as myogenin and HSP60 which promote growth and repair of skeletal muscle [47]. Thus, pyridoxine may have an overall protective effect against the development of sarcopenia and frailty in elderly populations, but more research is necessary. 

Marked cobalamin, or vitamin B12, deficiency can result in several neuromuscular symptoms, including muscle weakness, paresthesia, and numbness [48]. The link between B12 deficiency and frailty and sarcopenia has been studied previously. The same cross-sectional analysis of Dutch older adults found that greater intakes of vitamin B12 were partially correlated with higher scores on the chair rise test, but not on the SPPB [45]. Another cross-sectional study matching 66 non-sarcopenic older adults to previously recruited sarcopenic adults in a 1:1 ratio found that the sarcopenic group consumed significantly less vitamin B12 (22%) compared to the non-sarcopenic controls [49]. This same study also found that serum levels of vitamin B12 were 15% lower in the sarcopenic group than in the healthy controls [49]. Although around > 300 pg/mL of B12 is considered normal, this amount is considered borderline and may deem further testing [50]. One study found that when compared to elderly patients with levels of vitamin B12 > 400 pg/mL, patients with lower than 400 pg/mL had lower lean body mass, total skeletal mass, and skeletal muscle mass index [51]. The study also found a greater prevalence of sarcopenia and dynapenia in <400 pg/mL group, suggesting a link between vitamin B12 levels and the development of sarcopenia [51]. It is possible that B12 offers a protective effect against the development of sarcopenia and dynapenia, as many cross-sectional and prospective studies have found significant results linking them.

#### 2.2.4. Other Vitamins 

Other vitamins may play a potential role in the development of sarcopenia. Dietary carotenoids and vitamin A comprise an important component of the antioxidant defense system in humans [52,53]. This may implicate a role for them in the prevention of oxidative stress central to the pathogenesis of sarcopenia. In a cross-sectional study of men (*n* = 396) and women (*n* = 321) aged between 63 and 73 years, researchers found that higher intakes of β-carotene were associated with shorter 3 m walk and chair-rise times, but no such association was found in the men [37]. Another cross-sectional study conducted on 986 Italians (age 65 and older) initially found an association between levels of β-carotene intake and both knee extension strength and physical performance score (a composited score derived from ranking participants by walking speed, a chair-stand test, and s standing balance test), but the association with physical performance disappeared after adjusting for factors such as age, sex, smoking, and BMI [39]. The link between carotenoids and physical performance may be unclear, and furthermore influenced by several factors. One cross-sectional study of 2570 women aged 18–79 years attempted to study the relationship between the dietary intake of carotenoids and muscle mass in a sample stratified by age: they found that although intakes of total carotenoids were significantly associated with indices of fat-free mass and leg explosive power, the association was stronger in women under 65 years of age [54]. 

Although these studies measured dietary intake, serum carotenoid levels may serve as markers as well. Researchers assessed serum concentrations in 754 elderly women from the Women’s Health and Aging Studies I and II who were either classified as frail, prefrail, or non-frail based on Fried’s criteria [46,55]. The study found that the association between serum micronutrients and frailty was strongest for β-carotene, lutein/zeaxanthin, and total carotenoids after adjusting for age, sociodemographic status, smoking, and BMI [46]. Studies have shown that both dietary intakes and serum levels of vitamin A and carotenoids may play a role in the preservation of muscle mass, however randomized controlled trials and cohort studies are required to further explore this relationship. In addition, it should be considered whether it is the foods that supply these nutrients that derives the benefit rather than the specific nutrient.

Vitamin E’s potential role as a major antioxidant in the body has also been studied [56]. In a cross-sectional analysis of 227 older adults from the Maastricht Sarcopenia study, researchers discovered that sarcopenic older adults had a significantly lower intake of vitamin E when compared to non-sarcopenic older adults [44]. However, the same study also measured biochemical levels and found that there was no significant difference in serum α-tocopherol levels between the two groups [44]. In the cross-sectional analysis conducted in 986 Italians 65 years and older as described before, researchers found that plasma α-tocopherol levels were significantly correlated with knee extension and physical performance score, while plasma γ-tocopherol was only correlated with knee extension score, but dietary intake levels showed no significance [39]. Furthermore, researchers conducting a cross-sectional analysis of 1643 elderly adults from Spain found no significant association between the dietary intake of vitamin E and incident frailty, as defined by Fried’s criteria [40]. These results appear somewhat conflicting; however, there is error associated with dietary assessment and the serum marker is not perfectly correlated with dietary intake. Of note, however, a recent meta-analysis conducted on the effects of tocopherol supplementation on exercise performance and oxidative stress demonstrated no significant protection against either exercise-induced lipid peroxidation or muscle damage [57]. This study was not specific to elder populations, so the underlying effects may be different if stratified by age. However, supplementation with vitamin E has not proven effective even in elderly men, as discussed before [41]. In considering dietary intakes and serum micronutrient levels, vitamin E appears to have several conflicting results and there is a lack of convincing evidence that it is beneficially associated with the preservation of muscle mass and physical performance in elderly individuals. 

### 2.3. Minerals

#### 2.3.1. Calcium

Minerals are critical for both structural and regulatory functions within the body [58]. Calcium, for example, is the main regulatory signaling molecule for muscle fibers [59]. A role for calcium in sarcopenia has been suggested through their modulation of calpains, which are cysteine proteases responsible for the regulation of key processes in myogenesis [60]. A deficiency, therefore, could potentially lead to sarcopenic outcomes. 

A cross-sectional analysis of 396,283 participants through UK Biobank revealed that a higher intake of calcium was associated with lower odds of sarcopenia [61]. An analysis of the fourth Korea National Health and Nutrition Examination Survey (KNHANES) also found that daily calcium intake was positively correlated with appendicular skeletal muscle mass in 1339 older Korean adults [62]. Furthermore, after controlling for age, sex, BMI, total energy intake, and daily lifestyle factors, individuals in the highest tertile for daily calcium intake had an odds ratio for sarcopenia of 0.29 when compared to individuals in the lower tertile [62]. This appears to suggest a significant role for daily calcium intake in the prevention of sarcopenia. Another cross-sectional analysis, this one of the New Mexico Aging Process Study, measured nutrient intakes in response to gait speed. While slower men were shown to consume less calcium, women showed no such trend [36]. This suggests a potential sexual dimorphism for calcium intakes. In line with this hypothesis, one study examined the relationship between sarcopenic obesity and nutrition status in 1433 Korean adults 60 years and older, and found an association between calcium intake and body composition only in men, not women [38]. However, calcium intakes were found to be insufficient in all groups, meaning there may just not have been a functional difference between lower intake levels overall [38]. 

While some studies have shown a significant association between dietary calcium intake and sarcopenia, this is not the case for all. An analysis of the Maastricht Sarcopenia Study included 227 older adults and found that there was no significant difference in daily calcium intake between the sarcopenic and nonsarcopenic adults [44]. Another study matched 66 non-sarcopenic older adults to previously recruited sarcopenic older adults to study dietary intake differences, but found no significant difference between daily calcium intake in the two groups [49]. 

It should be noted that there is a startling lack of longitudinal cohort or randomized controlled trials studying the effects between calcium supplementation and the development of sarcopenia. Furthermore, the mechanism behind calcium’s effect on muscle mass warrants some attention. It may be pertinent to study the association between calpain levels and sarcopenia. The effect of daily calcium intake on sarcopenia is overall unclear and needs to be studied more in the future.

#### 2.3.2. Selenium

Selenium is an essential trace element which exerts most of its biological effects through selenocysteine in animals and bacteria. This amino acid is incorporated into selenoproteins through translational machinery, which generally function as oxidoreductase enzymes in various metabolic pathways protecting against oxidative damage [63]. Selenoprotein N was the first selenoprotein to be linked to congenital muscular dystrophies, but selenium deficiency has also been linked to muscle pain and weakness in previous studies [64,65,66]. Nutritional selenium deficiency has been described as well and corresponds to either nutritional muscular dystrophy or nutritional myotonic dystrophy [64]. As such, it is possible that selenium intake may influence sarcopenia development as well.

Several studies have examined differences in selenium intake between elderly subjects to determine selenium’s potential effect on sarcopenia. A case-control study matching 66 non-sarcopenic older adults to previously recruited adults with sarcopenia found a significantly lower intake of selenium in the sarcopenic group [49]. A cross-sectional analysis of 628 men and women in the Hertfordshire Cohort Study found that higher intakes of selenium were associated with better 3 m walk times, but not with other measure of physical performance, such as chair-rise times or balance [37]. No such association was found in men, suggesting a potential sexual dimorphism.

One study sought to identify whether differences in antioxidant intakes (such as selenium) differed between non-sarcopenic individuals and individuals with class I sarcopenia [67]. The study consisted of 16 elderly men and 34 elderly women and found no significant difference between the two groups in terms of selenium intake [67]. However, the sample size was rather low, and may have lacked sufficient power.

Since selenoenzymes may play a role in protecting muscle from oxidative damage, it is pertinent to examine serum levels of selenium as well. One cross-sectional study looked at the association between serum selenium and hand-grip strength in 676 women in the Women’s Health and Aging Study I and found that serum selenium was positively associated with grip strength [68]. A cross-sectional analysis of 891 men and women from the InCHIANTI study found that when compared to participants in the highest quartile, participants in the lowest quartile of plasma selenium had poorer hip strength, knee strength, and grip strength [69]. Serum selenium may thus be an appropriate marker for risk of muscle loss and sarcopenia. 

The supplementation of selenium has also been studied. In a randomized controlled trial, 443 elderly participants were either given a combination of 200 µg/day organic selenium yeast and 200 mg/day CoQ10, or a placebo for 4 years. Participants were evaluated for health-related quality of life (HR-QoL) using the Short Form-36, Cardiac Health Profile (CHP), and one item overall-quality of life (overall-QoL). A total of 206 participants were evaluated after 48 months, and, compared to the placebo group, the experimental group had fewer days in the hospital and also declined significantly less in terms of physical role performance, vitality, and physical component score [70]. The physical component score aggregates the domains of physical functioning, role limitations due to physical health problems, bodily pain, and general health from the HR-QoL into one composite score. 

Several studies have shown the association between plasma selenium and dietary selenium intake and physical performance and muscle mass. There has even been one clinical trial: although selenium was combined with CoQ10, there seemed to be a benefit. Though it is a trace element, it may be worth considering selenium for future studies involving sarcopenia. Furthermore, the specific mechanism behind selenium’s effect on muscle needs additional research; more randomized controlled trials could be integral to its success as a supplement or treatment. 

#### 2.3.3. Magnesium

Magnesium (Mg) is the second most abundant intracellular cation after potassium and is involved in over 600 enzymatic reactions, including energy metabolism and protein synthesis [71]. Animal studies have suggested that magnesium supplementation might improve exercise performance by enhancing glucose availability in the brain, muscle, and blood [72,73]. As a result, it is worth considering whether Mg intake plays a role in the physical activity and muscle mass of elderly individuals who are at risk of sarcopenia. 

A cross-sectional analysis of 396,283 participants using the UK Biobank revealed an inverse association between magnesium intake and sarcopenia [61]. Another cross-sectional analysis of the Maastricht Sarcopenia Study included 227 community-dwelling older adults and found that sarcopenic older adults had a significantly lower daily intake of Mg [44]. Interestingly, this analysis also studied serum magnesium between the two groups, and found no significant relationship between serum magnesium and sarcopenia [44]. A case control analysis matched 66 non-sarcopenic older adults to sarcopenic adults in a 1:1 ratio in order to examine dietary differences between them, and found that the sarcopenic group consumed significantly less magnesium than the non-sarcopenic controls [49]. Finally, a cross-sectional analysis of the InCHIANTI aging study examined 1138 elderly men and women and found that magnesium concentrations were significantly associated with physical performance, as evaluated by grip strength, lower-leg muscle power, knee extension torque, and ankle extension isometric strength [74].

A randomized controlled trial providing Mg supplementation to healthy elderly women in conjunction with a mild fitness program demonstrated significantly greater improvement in physical activity between the treated group and the control group, as assessed by Short Physical Performance Battery [75]. 

Dietary intake of magnesium appears to be related to physical activity and muscle performance in elderly individuals. The randomized controlled trial did conjoin magnesium supplementation with a regular fitness program, so it is uncertain whether magnesium supplementation may also be effective in sedentary individuals. Further research is necessary to understand this, but it seems that magnesium intake at least is an important consideration for the prevention of sarcopenia and the retention of muscle mass in elderly individuals. 

#### 2.3.4. Other Minerals

A few studies have shown associations between other minerals and sarcopenia as well. In a cross-sectional analysis of the UK Biobank, individuals who reported higher intakes of potassium were associated with lower odds of sarcopenia [61]. Potassium may play a role in preserving lean tissue mass through alkaline diets, which will be discussed in a later section. It is important to note that total body potassium is not a valid marker of potassium consumption in sarcopenic individuals, since muscle mass is the main potassium body store [76].

The effects of phosphorus on muscle mass have not been extensively studied. A study matching 66 non-sarcopenic older adults to adults previously recruited with sarcopenia in a 1:1 ratio found that the sarcopenic group consumed significantly less phosphorus in their diet [49]. However, a cross-sectional analysis of 7421 individuals from the National Health and Examination Survey sought to identify a link between serum phosphate levels and muscle strength; the study found a significant inverse association between phosphate and muscle strength in subjects > 65 years of age [77]. Further research is needed to consolidate a link between phosphorus and preservation of muscle mass. 

The effect of iron-deficiency on muscle growth is another poorly studied area. One randomized controlled trial in mice found that severe iron deficiency impaired general muscle growth, but that growth was restored shortly following iron repletion [78]. It is overall unclear how this study translates to humans. A cross-sectional analysis of 315 older adults in the New Mexico Aging Process Study measured gait speed between individuals and found that slower men consumed less iron per day, but slower women showed no such trend [36]. However, a longitudinal study of 698 elderly individuals from the InCHIANTI study sought to identify potential links between the serum concentrations of micronutrients and physical function decline, and found no association between baseline serum iron concentration and decline in physical function after a 3-month follow up [79].

Zinc deficiency has been shown to impair protein synthesis and accelerate protein degradation in rats [80], but it is unclear how this relationship translates to humans. In the same cross-sectional analysis of the New Mexico Aging Process Study, gait speed was not significantly associated with levels of zinc intake in men and women [36]. A cross-sectional analysis of 53 sarcopenic and 174 non-sarcopenic adults from the Maastricht Sarcopenia Study assessed dietary intake using a food frequency questionnaire and also found no significant difference in zinc intake between the two groups [44]. Finally, the same cross-sectional analysis matching 66 non-sarcopenic older adults to previously recruited adults with sarcopenia in a 1:1 ratio found no significant difference in zinc intake between the two groups as well [49]. Based on these studies, it appears that dietary zinc intake may not have a significant effect on preservation of muscle mass in older adults; however, additional prospective cohort studies and randomized controlled trials in this population are required to consolidate a relationship.

### 2.4. Antioxidants

#### Omega 3 Fatty Acids

Omega-3 (n-3) fatty acids are a class of long-chain fatty acids which have many beneficial biological effects [81,82]. They have long been hailed as having anti-inflammatory effects, and this may be particularly relevant in studies relating to sarcopenia. The most studied omega-3 fatty acids are eicosapentaenoic acid (EPA; 20: 5n-3) and docosahexaenoic acid (DHA; 22: 6n-3). The impact that omega-3 fatty acids may have on skeletal muscle systems has come to attention recently as researchers have theorized that these fatty acids may have a positive effect on muscle mass; this could be highly relevant in sarcopenic individuals. One cross-sectional study of 363 people aged 60 years and above assessed the relationship between dietary fish oil intake and frailty using the Edmonton Frail Scale (EFS) score, and found that oily fish intake had a positive effect only on the frailty status of younger subjects, but largely unaffected subjects over 70 years old [83].

Randomized controlled trials have been conducted to elucidate the impacts of omega-3 fatty acids in older individuals. After omega-3 supplementation (3.9 g/day) for 16 weeks, mixed muscle, mitochondria, and sarcoplasmic protein synthesis rates all increased in older adults before exercising, and mitochondria and myofibrillar protein synthesis rates increased post-exercise [84]. Studies have also demonstrated a protective effect against normal decline in muscle mass: 60 healthy older men and women were randomly assigned to receive either omega-3 fatty acid supplementation (1.86 g EPA and 1.50 g DHA) or corn oil therapy for 6 months, and, compared with the control group, the individuals receiving the n-3 supplementation had significantly increased thigh muscle volume, hand-grip strength, and 1-RM muscle strength (a composite score derived from leg press, chest press, knee extension, and knee flexion) [85]. Fish oil (FO) supplements (rich in n-3 fatty acids) were used in a study where 45 older women were randomly assigned to either a strength training only group for 90 days, a strength-training + FO group (0.4 g EPA and 0.3 g DHA) for 90 days, or a strength training + FO group (0.4 g EPA and 0.3 g DHA) for 150 days. Supplementation with FO resulted in significantly higher peak torque and rate of torque development levels in knee flexors and extensors, plantar and dorsiflexor muscles [86]. Fatty acids as incorporated in the diet have been studied as well: 63 healthy older women were randomized to either receive resistance training alone, resistance training + a healthy diet, or control. The “healthy diet” component emphasized keeping the n-6/n-3 polyunsaturated fatty acid ratio below 2, and found that whole-body lean mass increased significantly only in the group that received both resistance training and the healthy diet [87]. It is also possible that omega-3 fatty acids affect not the basal muscle synthesis rate, but only the rate in the fed state. In one study, 16 healthy older adults were randomly assigned to receive either a dietary supplement with 1.86 g EPA and 1.50 g DHA, or corn oil for 8 weeks. The rate of muscle protein synthesis was evaluated by using a phenylalanine tracer which was infused into the subjects, and this was measured before and after the administration of a hyperaminoacidemic–hyperinulinemic clamp, which provided amino acids and human insulin intravenously. The study concluded that, while omega-3 fatty acids had no effect on the basal muscle synthesis rate, they did significantly augment the hyperaminoacidemia–hyperinulinemia-induced increase in the muscle synthesis rate [88]. There may be gender differences in the reaction to FO supplementation. Twenty-seven men and 23 women were randomized to either n-3 fatty acid supplementation group (2.1 g EPA + 0.6 g DHA) or placebo for 18 weeks and found that there was a greater increase in muscle quality in women after exercise training in the n-3 supplementation group, but no such difference in men [89]. There was also an increase in maximal isometric torque in the supplementation group in women, but once again no such difference in men [89].

While several randomized controlled trials demonstrated significant effects between omega-3 supplementation, it is important to note the sample size in many of these trails. The largest of these randomized trials was the Multidomain Alzheimer Preventive Trial (MAPT) which was a 3-year multicenter trial with 1680 participants randomized to 4 parallel groups: a placebo only group, a low-dose omega-3 fatty acid supplementation only group + multidomain intervention (group sessions giving advice for physical activity/cognitive training/nutritional activity), a placebo + multidomain intervention group, and a placebo-only group [90]. The researchers observed no significant differences in chair-stand test score or hand-grip strength between any of the groups [90]. It is important to note, however, that the omega-3 supplementation in this study was overall lower (800 mg DHA and 225 mg EPA) than most of the other studies reviewed here. Interestingly, the 2015–2020 Dietary Guidelines for Americans (DGA) does not establish a Dietary Reference Intake for EPA and DHA, but do recommend that the general population consume about 8 oz of seafood/week, which provides about 250 mg EPA and DHA/day [91]. These recommendations may have to be reviewed as new evidence emerges from randomized controlled trials with sarcopenia in elderly individuals as the outcome.

### 2.5. Combination of Nutrients 

Numerous studies have evaluated the impact of supplementation with a combination of several nutrients with regard to muscle strength or mass or physical performance. The protein source has varied from a milk-based product to whey or branched chain amino acids (Table 3).

The most common supplement used was whey protein with a variety of other nutrients. In 380 sarcopenic individuals over the age of 65 years, 20 g of whey protein combined with 3 g of Leucine and 800 IU vitamin D vs. an isocaloric placebo for 13 weeks resulted in significant improvements in chair-stand test score with supplementation, as well as significant improvements in appendicular muscle mass when compared to the control [92]. In 130 individuals over the age of 65 with low muscle mass, supplementation with 22 g of whey protein and 100 IU vitamin D was compared to an isocaloric placebo for 12 weeks. In the supplemented group there was an increase in fat free mass, relative skeletal muscle mass, and grip strength as compared to the placebo group [93]. In a smaller study in 60 sarcopenic individuals aged 60 to 85, whey protein and vitamins D (702 IU) and E (109 mg) were provided in a supplement and compared to an isocaloric placebo after 6 months of supplementation. In this study, the relative skeletal mass index increased more in the supplemented group [98]. In a small metabolic study, 24 men were supplemented with 21 g of leucine enriched whey protein and vitamin D vs. a non-caloric placebo for 6 weeks and mixed muscle protein synthesis rate and lean mass in the leg increased in the supplemented group [94]. This may help in understanding how the protein supplement may prevent sarcopenia.

In a study of 107 individuals post hip fracture (age > 65 years), Ensure^®^ Plus Advance supplement (milk based) vs. a control group (no placebo), BMI, and appendicular lean mass decreased in the control group, whereas there was not a similar decrease in the supplemented group (*p* < 0.05) [96].

In a small study of 38 individuals over the age of 65 residing in a nursing home [95], L-Leucine (1.2 g) and 6 g of medium-chain triglycerides (MCT) was compared to L-Leucine and 6 g of long-chain triglycerides (LCT) and to an isocaloric placebo for 3 months. Supplementation with Leucine and MCT resulted in improved body weight, grip strength, and walking speed.

There were also two studies of specific amino acid supplementation. In 139 people over the age of 70 with sarcopenic obesity, a nutritional supplement provided essential amino acids (1.20 g leucine, 0.50 g lysine HCl, 0.33 g valine, 0.32 g isoleucine, 0.28 g threonine, 0.20 g phenylalanine, 0.17 g other) with 20 micrograms of vitamin D and tea fortified with 540 mg catechins. This supplement was given with or without exercise and compared to education alone or health education alone for 3 months. There was no significant difference in the skeletal muscle index, but the combined exercise and supplementation group was 3 times more likely to improve muscle strength assessed by walking speed than the education group [97]. In an 8-week study of 68 people over the age of 65 with low muscle mass and strength, a supplement consisting of 2500 mg of branched chain amino acids with 12.5 µg of vitamin D was compared to an isocaloric placebo. The supplemented group was found to have improved grip strength and larger calf circumference as compared to the placebo [99].

These studies evaluating combined nutrients were completed in various populations and used very different supplements. Although some of these combined supplements may have provided a benefit, it is unclear whether the benefit was equivalent to efforts leading to an improved diet in these individuals.

### 2.6. Food Groups

#### 2.6.1. Dairy

Milk proteins, such as casein and whey, are believed to be some of the highest-quality proteins. As a result, it is relevant to investigate the potential effects of dairy foods on sarcopenia. A cross-sectional analysis of 747 elderly individuals from the 2005 Korea National Health and Nutrition Examination Survey (KNHANES) found that the dietary intake of milk and milk-based products (milk, yogurt, ice cream) was associated with a significantly reduced level of functional disability in men, but not in women [100]. Although this study failed to find an association in women, another cross-sectional study evaluating the relationship between dairy intake and body composition in 1456 older women from Australia found that women in the highest tertile of dairy intake (milk, yogurt, and cheese) had significantly greater whole-body lean mass, appendicular skeletal muscle mass (ASMM), and hand-grip strength than women in the lowest tertile [101]. This study demonstrated a significant association between both physical performance and lean muscle mass and more frequent dairy intake. These conflicting results may relate to population ethnic differences, the fat content of the dairy consumed, or there may be a gender effect. 

A prospective cohort study conducted on 1871 older adults used the Fried criteria to assess incident frailty, and found that higher consumption of low-fat milk and yogurt was associated with a lower risk of frailty as well as a lower risk of slow walking speed after a 2-year follow up [102]. Functionally, this study appears to suggest that consumption of low-fat dairy products offers protection against the detrimental effects of sarcopenia. 

A study conducted in 2007 found that the consumption of fluid skim milk led to greater muscle protein accretion in young men after resistance exercise when compared to a soy-protein beverage [103]. This effect appears to translate to older populations as well: a randomized controlled trial found that supplementation with 210 g of ricotta cheese resulted in a significant relative change to ASMM when compared to the habitual diet control group after 12 weeks [104]. In fact, ASMM actually increased in the group supplemented with ricotta cheese, while it decreased in the control [104].

Recent reports appear to suggest that increased dairy consumption (cheese, low-fat dairy, and milk products) imparts some protective effects against sarcopenia and frailty, but further randomized controlled trials are necessary to confirm these results, especially in older adults. Although the current recommendations to prevent frailty include protein supplementation, it may be prudent to determine whether the form or type of protein is important. Furthermore, whether the beneficial effects of dairy consumption in older adults is unique to the combination of nutrients with the protein provided in dairy deserves further study. 

#### 2.6.2. Tea

Oxidative stress has been suggested to play a role in sarcopenia [105]. Green tea polyphenols and catechins (TC) have been shown to exhibit anti-oxidant effects, which may be relevant in the context of sarcopenia for older individuals [106,107]. In senescence-accelerated prone mice, TC supplementation combined with habitual exercise has been shown to decrease oxidative stress and exhibit a protective effect on endurance capacity when compared to mice who did not receive TC [108]. There has only been one human study, conducted on 128 elderly Japanese women: individuals who received both catechins and exercise training improved the most in terms of usual walking speed and leg muscle mass [107]. However, individuals receiving only TC supplementation were not significantly different from the control group, suggesting the effects of TC supplementation may only amplify the normal response exhibited from exercising [107]. Regardless, further research is required to understand the effects of TC supplementation.

#### 2.6.3. Fruits and Vegetables 

Although this review has focused most of its attention on individual micronutrients and macronutrients, and their respective studied effects on sarcopenia and frailty, it may be relevant to discuss certain food groups as well, and their imparted effects. A cross-sectional study of 823 men and 1089 women over 65 years old found that the dietary intake of fruits, vegetables, and both fruits and vegetables was significantly inversely associated with sarcopenia, defined as proposed by Newman et al. [109,110]. In another cross-sectional analysis conducted in 2983 men and women between the ages of 59 and 73 from the Hertfordshire cohort, researchers found a significant positive association between weekly intake of both fruits and vegetables and grip strength in women, but only between fruits and grip strength in men [111]. In a cross-sectional analysis of adults aged 60 years or older from the NHANES cohort, 2132 adults had recorded gait speed data and 1392 had recorded knee extensor power data [112]. Researchers measured diet quality using the USDA’s Healthy Eating Index-2005 (HEI-2005) and found that higher HEI-2005 scores for total fruit, whole fruit, and dark green and orange vegetables were associated with faster gait speed [112]. However, only higher HEI-2005 scores for total fruit and whole fruit were associated with greater knee extension power [112]. Another key factor to consider is diversity within the diet: consuming a variety of foods (especially fruits and vegetables) is recommended by the Dietary Guidelines for Americans [113]. A cross-sectional study conducted in 36 men and 62 women aged 72 to 98 years sought to assess the relationship between the variety in the diets of participants and various markers of nutritional status [114]. The researchers found that the 3-day fruit and vegetable variety score was positively associated with BMI, mid-arm circumference, and mid-arm muscle area in women, but not men [114]. However, since there were about only half as many men as there were women in this study, there may not have been enough power to establish an association in men. 

Dietary intake may also be associated with the preservation of muscle mass. A longitudinal cohort study consisting of 575 older Japanese women found that more frequent intake of green and yellow vegetables at baseline was associated with less age-related knee extension strength decline after a 4-year follow up [115]. Another longitudinal study conducted in 2948 Chinese men (n = 1449) and women (n = 1499) aged 65 years and older found that fruit and vegetable consumption at baseline was associated with lower odds of sarcopenia in men only, but no association between dietary patterns and incident sarcopenia was seen after a 4-year follow up in both sexes [116]. Incident frailty has also been studied as an outcome. A meta-analysis of three independent cohort studies on community-dwelling older men and women found higher portions of both and fruit vegetable intake to be associated with a lower risk of frailty (as defined by the Fried criteria) after a mean of 2.5-year follow up [117]. 

Many cross-sectional and cohort studies present evidence supporting the role of fruits and vegetables in preserving muscle mass and physical function, as well as preventing sarcopenia and frailty. The mechanism behind which this diet exerts its protective effects is unknown, but since metabolic acidosis has been linked to muscle wasting in other diseases [118], one hypothesis is that alkaline diets may preserve muscle mass. A longitudinal cohort study sought to assess the effects of an alkaline diet on muscle mass in 384 men and women above 65 years of age, and found that urinary potassium was significantly positively associated with the percentage of lean body mass at baseline, but not with a 3-year change [119]. The study concluded that foods rich in potassium, that are associated with lower dietary acid load, may assist in preserving muscle mass [119]. In support of this hypothesis, a randomized controlled trial with 162 healthy men and women aged 50 and older found that bicarbonate supplementation improved leg press power in women [120]. A cross-sectional study on 2176 Japanese women aged 65–94 years found that a higher dietary acid load was positively associated with frailty [121], but once again did not explicitly demonstrate that an alkaline diet is necessarily superior. Further research is needed to conclude whether fruits and vegetables exert their effects through their alkaline properties, antioxidant properties, some other component, or if intake is simply a marker of a healthy diet. However, results of several studies support the role of fruits and vegetables overall in preserving muscle mass and preventing sarcopenia and frailty. Randomized controlled trails would be useful in further consolidating this relationship and perhaps helping to find a mechanism.

### 2.7. Dietary Patterns 

#### Mediterranean Diet

The Mediterranean Diet is a healthy alternative eating pattern characterized by a high intake of whole grains, vegetables, fruits, fish, and nuts; a moderate consumption of alcohol and olive oil; and a low consumption of red meat [122]. We have seen in this review that several of the nutrient components of the Mediterranean Diet may exert protective effects against sarcopenia and frailty, as well as play a role in preserving muscle mass and physical ability. A cross-sectional analysis amongst 2791 older adults from the NHANES cohort and 1786 older adults from the Israeli National Health and Nutrition Survey found that higher adherence to the Mediterranean Diet was associated with faster walking speed and fewer physical disabilities [123]. Another cross-sectional study conducted in 300 elderly men and women from Iran found that higher adherence to the Mediterranean Diet was associated with lower odds of sarcopenia, although adherence to the Western dietary pattern was not significantly associated with sarcopenia [124]. In the UK, a cross-sectional study conducted in 2570 women between the ages of 18–79 from the TwinsUK study found that higher adherence to the Mediterranean Diet was positively associated with fat-free mass percentage, but not with grip strength [125].

A longitudinal study of 2225 individuals 70 years and older from the American Health ABC cohort found that adherence to the Mediterranean Diet was associated with less decline in 20 m walk speeds after an 8-year follow up, even after adjusting for age, race, sex, and several other factors [126]. Although this points to a link between the diet and physical performance, this relationship may not extend to protection against overall sarcopenia and frailty. Another prospective cohort study of 2724 older Chinese men and women found no significant association between Mediterranean Diet adherence and incident frailty after a 4-year follow up [127]. In contrast, a similar cohort study conducted in 1815 elderly individuals from Spain found that higher adherence to the Mediterranean Diet was associated with reduced risk of slow walking, weight loss, and incident frailty after a mean follow up of 3.5 years [128]. These results were also replicated in Italian subjects: a cohort study of 690 adults aged 65 years and above from the InCHIANTI study found that a higher adherence to the Mediterranean Diet was associated with a lower risk of low physical activity, low walking speed, and development of frailty, but not with poor muscle strength [129]. A cohort study of 554 women aged 65–72 years from the Finnish Osteoporosis Risk Factor and Prevention-Fracture Prevention Study (OSTPRE-FPS) found that women with higher adherence to the Mediterranean Diet had significantly faster walking speed and greater lower body muscle quality after a 3-year follow up [130].

In regard to sarcopenia specifically, a longitudinal cohort study of 2948 Chinese men and women aged 65 years and up found no association between Mediterranean Dietary adherence and incident sarcopenia after a 4-year follow up in both sexes [116].

A longitudinal study conducted on 1410 French participants from the Three-City Bordeaux cohort found no association between Mediterranean Diet adherence and the risk of incident disability over 5 years in men (n = 527), but did find a relative risk reduction for incident frailty in women (n = 883) [131], suggesting the relationship may have some gender specificity. Interestingly, another longitudinal study using 560 non-frail individuals from the same cohort found that higher adherence to the Mediterranean Diet was associated with significant frailty risk reduction in all participants after a 2-year follow up [132].

A recent meta-analysis included 12 studies with a total of 20,518 subjects and found that higher adherence to a Mediterranean Diet was inversely associated with both frailty and functional disability, but was unable to draw conclusions on its relationship to sarcopenia due to variations in study design [133].

Although adherence to the Mediterranean Diet may be beneficial to prevent frailty and sarcopenia in older adults, there has been great variability in the populations studied. The ethnic differences in populations and how they might modify the components of the Mediterranean Diet could have an impact on the results. Furthermore, more studies need to evaluate the impact of the Mediterranean Diet on incident sarcopenia, while there is mounting evidence to support the hypothesis that the Mediterranean Diet preserves muscle mass and some physical function in aging populations.

### 2.8. Assessment of Dietary Factors 

Considering the various potential effects that diet may have on sarcopenia, some assessment of nutritional status should be standardized for elderly patients to evaluate risks. The Mini Nutritional Assessment (MNA) and the Mini Nutritional Assessment Short Form (MNA-SF) are two such screening tools which may be used to identify malnutrition and evaluate nutritional status in elderly patients during clinical visits [134,135]. The forms can be conducted by relatively untrained personnel, and have been shown to have high sensitivity and specificity levels for malnutrition as well as frailty [136,137]. If there is even insufficient time for the MNA, then an alternative may be simply asking patients about their dietary intake on a typical day and watching for those that have no protein sources or fruits and vegetables in their diets. Furthermore, it is critical to be mindful of any issues of food insecurity: the USDA 18-item Households Food Security Survey Module can be used, although a “Short-Form” 6-item module is also available if time constraints are an issue [138]. In terms of laboratory tests for nutritional assessment, typical signs of a poor diet may include anemia [139], B-12 deficiency, prealbumin levels [140], or abnormal creatine phosphokinase (CPK) levels, which have historically been thought to signal muscle wasting, although there has been some controversy in recent literature [141]. In addition, vitamin D deficiency or insufficiency can be easily detected with a measurement of 25(OH)D and treated with supplementation. 

If there is an issue with food security or if a patient is unable to prepare a meal or gain access to necessary foods, there are several programs available for support. These include the Supplemental Nutrition Assistance Program (SNAP), Meals on Wheels, the Office for Aging Luncheon Sites, Senior Farmers Market Nutrition Program (SFMNP), and the Commodity Supplemental Food Program (CSFP), as well as other community-driven pantries and programs. Nutritionists and clinicians should be well versed in available opportunities to best support elderly patients who may be concerned about their dietary intake, since their diet could have major implications for their physical function and well-being. 

## 3. Discussion and Conclusions

Many factors, such as oxidative stress and inflammation, may play a role in the pathogenesis of sarcopenia. Overall muscle health is important for the functionality and independence of older adults, and certain nutrients as well as dietary patterns have been shown to offer protective effects against normal strength and functional declines associated with aging. Certain proposed mechanisms of effect have been discussed as well, but these largely require more research overall.

Protein and vitamin D are amongst the most studied of these nutrients, and various randomized controlled trials appear to suggest a critical role for the dietary intake of protein in preventing sarcopenia and muscle loss. However, further studies are still necessary to determine what amount and source of dietary protein may offer the greatest benefit. There are some promising data regarding the role of vitamin D and sarcopenia, but it is unclear whether the dose, frequency of dose, or length of treatment impacts the efficacy of vitamin D on improving muscle mass or function.

In terms of micronutrients, selenium and magnesium have been studied in both randomized controlled trials as supplements and in observational studies of the diet, and they appear to have a potential association with physical activity and muscle performance in older individuals. These micronutrients are worth further exploring and may warrant attention as important mediators in the development of sarcopenia. Meanwhile, omega-3 fatty acids have repeatedly demonstrated value in preserving muscle mass and protecting against normal decline in elderly individuals, both in randomized controlled trials and cohort analyses. While there is no established Dietary Reference Intake for EPA or DHA, this recommendation may have to be reviewed with respect to older individuals. However, dosage amounts vary in controlled trials, and it is still unclear which dosage is most effective. Further research is required, but omega-3 fatty acids appear to have potential as anti-sarcopenic compounds.

The Mediterranean Diet is characterized by several healthy eating behaviors, and one such behavior includes a high consumption of fruits and vegetables. In terms of both adherence to the Mediterranean Diet and the consumption of fruits and vegetables, all but one study demonstrated some association with either physical performance, protection against muscle wasting, or the development of sarcopenia and frailty. However, there are no randomized controlled trials, and although difficult to perform these could assist in determining a relationship between the Mediterranean Diet, intake of fruits and vegetables, and sarcopenia. Nevertheless, meta-analyses found associations between both the Mediterranean Diet and incident frailty as well as between fruit and vegetable consumption and incident frailty.

It is important to consider is the great variability in results obtained from different analyses. Many studies appear to suggest a potential sexual dimorphism between men and women, and this remains to be further studied. Cultural and ethnic differences in dietary consumption and method of preferred food preparation may further impact various results: cohort analyses in different countries did not always produce similar results. These are factors that must be considered as well when making dietary recommendations and must be incorporated into future studies. In addition, it would be important to consider standardizing some of the outcomes that are used in future studies.

Genetic variation may play a role in explaining some of this variability: the field of nutrigenetics focuses on the effects of such variation on dietary responses, as well as the role of bioactive nutrients in gene expression [142,143]. Sarcopenia’s genetic basis has drawn attention, and various genes involved in physical function and muscle mass have been identified [144]. The fundamental goal within nutrigenetics is to connect the dietary intake to the genomic status of the individual; however, to accomplish such a task would require a comprehensive understanding of both the genetic influences behind a certain pathology, as well as the mechanisms by which various nutrients affect gene expression in relevant metabolic pathways. Significant research is thus required regarding the molecular effects of bioactive nutrients; nevertheless, this may be a potential way to explain the great variability between results.

After a review of the existing evidence, it is possible that a well-planned diet may work just as effectively, or possibly better, than individual nutrient supplements in preserving muscle mass and physical function in elderly individuals. Supplements, such as protein with certain key micronutrients, may be most useful to those individuals unable to follow a healthy diet due to factors such as cognitive decline or an inability to prepare a meal. Considering the current literature, it may be pertinent to clinically assess nutritional status in elderly individuals who are at risk for sarcopenia and frailty through surveys, weight assessment, and selected bloodwork, since an adequate nutritional profile appears to assist in sustaining muscle mass and maintaining levels of physical function.

## Figures and Tables

**Table 1 nutrients-12-01755-t001:** Randomized controlled trials of protein supplementation and sarcopenia.

Author	Population	Intervention	Results of Supplemental Protein vs. Placebo	With Exercise?
Björkman 2010 [15]	*n* = 218, age 75–96 years, either low hand-grip strength (men ≤ 30.0 kg, women ≤ 20.0 kg), slow habitual gait speed (≤0.80 m/s), or cRi-SMI Z-score ≤ −2.	Nutritional supplement: enriched milk protein (40 g). Groups: supplement group vs. isocaloric placebo for 12 months.	Nonsignificant differential impact on physical performance or attenuation of muscle loss.	Yes, participants given a training program for 12 months.
Tieland 2012 [16]	*n* = 62, age ≥ 65 years, met Fried criteria for either frailty or pre-frailty.	Nutritional supplement: 15 g protein beverage supplemented by milk protein concentrate. Groups: Supplemented group vs. isocaloric placebo group for 24 weeks.	Significant differential increase in lean body mass in intervention group, but no significant differential change in strength and physical performance between groups.	Yes, both groups received resistance-type exercise training.
Kang 2019 [17]	*n* = 115, age ≥ 60 years, met at least 2 of 5 components of Fried’s physical frailty.	Nutritional supplementation: Nutrasumma whey protein. Groups: Supplemented group vs. control for 12 weeks.	Significant differential increase in hand-grip strength, gait speed, and chair-stand time.	Yes.
Park 2018 [18]	*n* = 120, age 70–85 years, met ≥1 of modified CHS frailty criteria, and MNA score ≤ 23.5.	Groups: 0.8 g whey protein/kg body weight supplementation vs. 1.2 g whey protein/kg body weight vs. 1.5 g protein/kg body weight 1:1:1 ratio for 12 weeks.	Significant differential increase in appendicular skeletal muscle mass (ASM), skeletal muscle mass index (SMI), and gait speed in 1.5 g protein/kg body weight group vs. 0.8 g protein/kg body weight group. No significant differential increase between 0.8 g protein/kg body weight and 1.2 g protein/kg body weight groups.	No, participants were told to maintain usual physical activity.
Yamada 2019 [19]	*n* = 112, age ≥ 65 years, met criteria for either sarcopenia or dynapenia.	Nutritional supplementation: 10.0 g whey protein + 800 IU Vitamin D. Groups: Exercise + nutritional supplement vs. exercise alone vs. nutritional supplement alone vs. control 1:1:1:1 ratio for 12 weeks.	Significant differential improvement in combined group’s appendicular muscle mass for sarcopenic adults, but not in adults with low physical function and normal muscle mass.	Yes, resistance program provided and part of randomization.
Amasene 2019 [20]	*n* = 41 (28 analyzed), age ≥ 70 years, met criteria for sarcopenia.	Nutritional supplement: 20 g whey protein isolate enriched with 3 g leucine. Groups: Supplemented group vs. isocaloric placebo for 12 weeks.	No significant differential improvement in physical function or muscle mass.	Yes, both groups followed a supervised resistance training program.
Chalé 2013 [21]	*n* = 80, age 70–85 years, SPPB score ≤ 10.	Nutritional Supplement: 40 g/day whey protein concentrate (20 g protein, 25 g maltodextrin). Groups: Supplemented group vs. isocaloric control.	No significant differential improvement between the supplemented group and control group.	Yes, resistance training protocol 3×/week for 6 months.
Daly 2014 [22]	*n* = 100, age 60–90 years, healthy.	Nutritional supplementation: 160 g cooked lean red meat. Groups: Red meat + resistance training vs. resistance training alone.	Red meat + PRT group experienced significantly greater gains in total body lean tissue mass, leg lean tissue mass, and muscle strength than control.	Yes, 45–60 min resistance training program for all participants.

CHS = Cardiovascular Health Study, SPPB = Short Physical Performance Battery, PRT = Progressive Resistance Training.

**Table 2 nutrients-12-01755-t002:** Randomized controlled trials of vitamin D supplementation and sarcopenia.

Author	Population	Intervention	Results of Supplemental Vitamin D vs. Placebo	With Exercise?
Hajj 2019 [30]	*n* = 128 (115 completed), average age = 73.31 years, met criteria for sarcopenia and vitamin D deficiency (<20 ng/mL).	Nutritional supplement: 10,000 IU cholecalciferol tablet. Nutritional supplement vs. placebo 3 times/week 6 months.	Significant differential improvement in appendicular skeletal muscle mass but not in hand-grip strength. Significant interaction between vitamin D and obesity also shown.	No.
Suebthawinkul 2018 [31]	*n* = 88, age 45–60 years, postmenopausal women with vitD deficiency (<20 ng/mL).	Nutritional supplement: 40,000 IUvitD/week. Groups: Nutritional supplement vs. placebo in 1:1 ratio for 12 weeks.	No significant differential improvement in muscle strength, muscle mass, or muscle CSA.	No, participants were asked to maintain normal physical activity.
Cangussu 2015 [32]	*n* = 160, age 50–65 years, postmenopausal women.	Nutritional supplement: vitamin D_3_ supplementation 1000 IU/day orally. Groups: Supplement group vs. placebo group in 1:1 ratio for 9 months.	Significant differential increase in muscle strength of lower limbs in intervention group.	No.

**Table 3 nutrients-12-01755-t003:** Randomized controlled trials of combination therapy and sarcopenia.

Author	Population	Intervention	Results of Supplemental Protein + Vitamin D vs. Placebo	With Exercise?
Bauer 2015 [92]	*n* = 380 (297 completed), age ≥ 65 years, with class I or II sarcopenia.	20 g whey protein, 3 g leucine, 800 IU VitD supplement vs. isocaloric placebo supplement for 13 weeks.	Nonsignificant differential increase in SPPB performance and hand-grip strength in both groups. Significant increase in chair-stand test performance in experimental group vs. placebo group. Experimental group also gained more appendicular muscle mass than control.	No.
Rondanelli 2016 [93]	*n* = 130, age ≥ 65 years, relative muscle mass Z score ≤ 2.	Nutritional supplement: 22 g whey protein, 100 IU vitD. Groups: Supplement group vs. isocaloric placebo for 12 weeks.	Significant differential increase in fat-free mass, relative skeletal muscle mass, and hand-grip strength in the intervention group.	Yes, both groups received physical activity training.
Chanet 2017 [94]	*n* = 24, age ≥ 65 years, men with BMI between 20 and 30.	Nutritional supplement: 21 g leucine-enriched whey protein + 800 IU vitD_3_. Groups: Nutritional supplement vs. noncaloric placebo before breakfast for 6 weeks.	Significant differential improvement in mixed muscle protein fixed synthesis rate (FSR) as well as appendicular lean mass growth in the intervention group, mainly as lean leg mass.	No.
Abe 2016 [95]	*n* = 38, age ≥ 65 years, resided in nursing home and required special care from helper.	Nutritional supplement: L-leucine (1.2 g), vitD_3_ (20 ug). Groups: Leucine supplement (LD) + 6 g Medium Chain Triglycerides (MCTs) vs. Leucine supplement + 6 g Long Chain Triglycerides (LCTs) vs. isocaloric placebo for 3 months.	Significant differential improvement in bodyweight, grip strength, and walking speed in the LD + MCT group only.	No.
Malafarina 2017 [96]	*n* = 107 (92 completed), age ≥ 65, with hip fractures.	Nutritional supplement: Ensure^®^ Plus Advance containing 20.02 g milk protein, Calcium β-hydroxy-β-methylbutyrate (CaHMB) 1.54 g, 499.4IU vitD, 499.4 mg Ca. Groups: Nutritional supplement group vs. control group (no placebo supplement) until discharge from rehabilitation center.	Significant differential in intervention group; appendicular lean mass and BMI were stable while in the control group, these factors decreased.	No, but patients were undergoing rehabilitation for hip fracture.
Kim 2016 [97]	*n* = 139, age ≥ 70 years, met criteria for sarcopenic obesity.	Nutritional supplement: EAA supplement + 20 ug vitD + tea fortified with 540 mg catechins. Groups: Exercise + nutrition vs. exercise alone vs. nutrition alone vs. health education group for 3 months.	Significant differential reduction in trunk fat and total body mass in thd Exercise + nutrition group against the health education group. No significant difference in SMI, but Ex + N group 3 times more likely to improve muscle strength than HE group.	Yes, aerobic and resistance training both included for Ex + N and Ex alone groups.

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
