# Peer review of "Nutrition and Sarcopenia—What Do We Know?"

_nutrients, 2020, doi:10.3390/nu12061755_

Round 1

Reviewer 1 Report

Thank you very much for your paper. Really an hard job reviewing a such amount of literature. I really appreciated your job. For sure nutrition plays a key role for sarcopenia, even if as you stated, often data on single micronutrients are not so strong, Evenb because very often trials include a physical activity protocol, that may overhelmed effects of nutrition. Nevertheless. your paper present a clear "state of the art" on this topic

Author Response

Response to reviewers:

We would like to thank the reviewers for their positive comments about the quality of the manuscript on this important topic.

Reviewer 1- spell check the document

Spellchecked/grammar checked and made minor corrections throughout, including spacing.

We have also fixed the citation issue in Bauer citation.

Reviewer 2 Report

The article entitled "Nutrition and Sarcopenia - What do we Know?" good quality and well written. The topic of the article is relevant and has been approached in a systematic way.
I believe that the introduction could be improved. The risk factors for this condition could be described, as well as diagnostic methods. In lines 280 to 282 the types of Sarcopenia are described, but I think this distinction should appear in the introduction of the article.
Two other aspects that could greatly enrich this article may be added. An sub-chapter dedicated to the assessment of nutritional status and how maintaining good nutritional status may be important to prevent Sarcopenia and another subchapter that could be very relevant to enrich the article would be one dedicated to nutrigenetics, since Sarcopenia is also influenced by genetics. Alternatively, this approach could be included in the introduction.

Author Response

Reviewer 2:

The article entitled "Nutrition and Sarcopenia - What do we Know?" good quality and well written. The topic of the article is relevant and has been approached in a systematic way.

Thank you for your comments.

I believe that the introduction could be improved. The risk factors for this condition could be described, as well as diagnostic methods. In lines 280 to 282 the types of Sarcopenia are described, but I think this distinction should appear in the introduction of the article.

We have moved the definition from lines 280 noted above to the introduction and have added sentences regarding risk factors and diagnostic methods (lines 31-39).

Two other aspects that could greatly enrich this article may be added. A sub-chapter dedicated to the assessment of nutritional status and how maintaining good nutritional status may be important to prevent Sarcopenia

We thank you for this suggestion and we have added a subchapter on assessment of nutritional status on lines 647-671.  We have also added the importance of this in to the discussion (lines 727-731).

another subchapter that could be very relevant to enrich the article would be one dedicated to nutrigenetics, since Sarcopenia is also influenced by genetics. Alternatively, this approach could be included in the introduction.

Thank you for this suggestion, we decided the paper would flow better if we added the section on nutrigenetics in the discussion section- it is found on lines 713 to 722.